# Dynamic *In Vitro* PK/PD Infection Models for the Development and Optimisation of Antimicrobial Regimens: A Narrative Review

**DOI:** 10.3390/antibiotics13121201

**Published:** 2024-12-10

**Authors:** Yalew M. Wale, Jason A. Roberts, Fekade B. Sime

**Affiliations:** 1Centre for Clinical Research (UQCCR), Faculty of Medicine, The University of Queensland, Brisbane, QLD 4029, Australia; 2Department of Pharmacy, College of Health Sciences, Debre Markos University, Debre Markos P.O. Box 269, Ethiopia; 3Departments of Pharmacy and Intensive Care Medicine, Royal Brisbane and Women’s Hospital, Brisbane, QLD 4006, Australia; 4Division of Anesthesia Critical Care and Emergency and Pain Medicine, Nimes University Hospital, University of Montpellier, UR UM 103, 34090 Nimes, France; 5Herston Infectious Diseases Institute (HeIDI), Metro North Health, Brisbane, QLD 4006, Australia

**Keywords:** dynamic *in vitro* infection models, antimicrobials development, optimisation of antimicrobial regimens

## Abstract

The antimicrobial concentration–time profile in humans affects antimicrobial activity, and as such, it is critical for preclinical infection models to simulate human-like dynamic concentration–time profiles for maximal translatability. This review discusses the setup, principle, and application of various dynamic *in vitro* PK/PD infection models commonly used in the development and optimisation of antimicrobial treatment regimens. It covers the commonly used dynamic *in vitro* infection models, including the one-compartment model, hollow fibre infection model, biofilm model, bladder infection model, and aspergillus infection model. It summarises the mathematical methods for the simulation of the pharmacokinetic profile of single or multiple antimicrobials when using the serial or parallel configurations of *in vitro* systems. Dynamic *in vitro* models offer reliable pharmacokinetic/pharmacodynamic data to help define the initial dosing regimens of new antimicrobials that can be developed further in clinical trials. They can also help in the optimisation of dosing regimens for existing antimicrobials, especially in the presence of emerging antimicrobial resistance. In conclusion, dynamic *in vitro* infection models replicate the interactions that occur between microorganisms and dynamic antimicrobial exposures in the human body to generate data highly predictive of the clinical efficacy. They are particularly useful for the development new treatment strategies against antimicrobial-resistant pathogens.

## 1. Introduction

The effectiveness of antimicrobials for the treatment and prevention of infections is informed by preclinical and clinical studies. Clinical trials are considered the gold standard for assessing the effectiveness and safety of antimicrobials [1,2]. However, prior to clinical studies, adequate safety and efficacy of new compounds should be established using preclinical models [3,4]. Animal-based models are employed for evaluating the safety and pharmacodynamics of antimicrobials [5,6,7]. These studies can measure the complex interactions between the immune system, pathogens, and antimicrobials [8,9]. Animal models offer an invaluable tool for the preclinical development of an antimicrobial; however, animal models have important limitations [10,11], including the ethical concerns regarding animal rights and welfare [12,13]. Moreover, the follow-up period before determining treatment outcomes (e.g., survival) in animal models does not sufficiently align with the disease process in humans [11]. For instance, it is difficult to monitor the emergence of resistant bacteria during the course of treatment in the most commonly used rodent animal models, often due to the death of the animals [14] and the inability to sustain the model over a clinically relevant duration. Consequently, various *in vitro* infection models are used to study the effect of antimicrobial exposure on bacterial killing and the emergence of resistance for different dosing regimens and course durations.

Although the *in vitro* infection models do not fully replicate the interactions of the host, antimicrobials, and microorganisms [15,16], they provide crucial preclinical insights into the direct actions of antimicrobials against pathogenic organisms without the involvement of the immune system. Hence, *in vitro* infection models contribute significant insights for the optimisation of antimicrobial dosing regimens. These models are categorised into static and dynamic models based on whether the antimicrobial concentration remains constant or changes during the study period [17].

In static *in vitro* infection models, the pathogenic organisms are exposed to a fixed concentration of the antimicrobial/s [18], unlike dynamic *in vitro* infection models, in which the concentration of the antimicrobial/s changes during the experimental period. Hence, the dynamic *in vitro* models mimic the fluctuation in antimicrobial exposure in the human body [15,19]. The dynamic antimicrobial exposure in dynamic *in vitro* models is achieved by a controlled flow of drug-free growth media at a rate mimicking the pharmacokinetics of the antimicrobial/s [20,21].

The simulation of humanised pharmacokinetic profile of the antimicrobials helps to characterise their pharmacokinetic/pharmacodynamic (PK/PD) indices, which serve as surrogate indicators of efficacy in describing dose–exposure–response relationships. The common indices include the ratio of maximum concentration (C_max_) to the minimum inhibitory concentration (MIC) (C_max/_MIC), the fraction of dosing interval for which the free antimicrobial concentration is above the MIC (*f*T_>MIC_), and the ratio of the area under the curve (AUC) of concentration-vs.-time plot to the MIC (AUC/MIC) [14,22,23,24]. The determination of the PK/PD index in these models helps optimise the dose and treatment schedule of both new and commercially existing antimicrobials [25,26]. The optimised dose and dosing schedule of antimicrobials derived from dynamic *in vitro* infection models is often predictive of clinically effective dosing regimens with a high rate of accuracy for translation into clinical settings [27]; this is often prospectively validated in clinical studies prior to implementation in routine clinical practice [28,29]. Dynamic *in vitro* infections models are, therefore, invaluable tools both in the development and post-marketing optimisation of antimicrobial therapy.

While there has been an increasing recognition of the utility of dynamic *in vitro* infection models in both academia and industry, the wider application of these models in antimicrobial optimisation is relatively limited. This is in part due the technical intricacies of the model and the lack of detailed published guidance in the model setup and pharmacokinetic simulation approaches. Therefore, the purpose of this review is to discuss the model setup, mathematical simulation approaches, and application of dynamic *in vitro* PK/PD infection models used in antimicrobial development and optimisation. Static *in vitro* infection models and dynamic flow *in vitro* models that do not simulate or are not commonly used to simulate humanised pharmacokinetics and animal models are beyond the scope of this review paper.

## 2. Dynamic *In Vitro* Infection Model Setups and Procedures

The dynamic *in vitro* infection models include the one-compartment dynamic *in vitro* model, hollow fibre infection model (HFIM), dynamic *in vitro* biofilm model, dynamic *in vitro* bladder infection model, and dynamic *in vitro* aspergillus infection model.

### 2.1. One-Compartment Dynamic In Vitro Model

The setup in the one-compartment infection model consists of a central reservoir, diluent reservoir, peristaltic pumps, and waste reservoir, as illustrated in Figure 1 [17,28,30].

The central reservoir represents the infection compartment into which the antimicrobial and microbial inoculum are combined. It is placed on a magnetic stirrer to ensure continuous mixing of the culture and antimicrobials [30]. The change in antimicrobial concentration within the central compartment is achieved by a continuous supply of drug-free nutrient broth at a rate that replicates the desired pharmacokinetics [15,18]. Growth media is pumped from the diluent reservoir to the central reservoir via substitution (stepwise dilution) [30,31], with an equal amount of fluid being removed to keep the volume constant [31]. The antimicrobial is added to the central reservoir by infusion. For accurate simulation of the desired concentration–time profile, it is critically important to precisely calibrate the flow rate of the pumps and set the flow rate of media based on the half-life (clearance) of the drug that is most representative of the study population for which the antimicrobial indication is being investigated. Accurate simulation of the desired pharmacokinetic profile should be confirmed by measuring the concentration of the antimicrobial in a serial of samples collected from the central reservoir using validated assay methods.

When simulating the pharmacokinetics of oral regimens, an additional reservoir (antimicrobial reservoir) is added between the diluent reservoir and the central reservoir (Figure 2). The antimicrobial solution is administered to its reservoir (Figure 2) to simulate the gastrointestinal absorption of the drugs [28] after oral administration.

In the simplest setup, the central reservoir in the one-compartment model can be an open system [17]. In advanced versions of the setup, this open *in vitro* system is replaced by a closed system to secure the biosafety of the laboratory environment and the technical individual working with the equipment [17,32]. A sealed one-compartment *in vitro* infection model requires a central reservoir with a tight cap, ports for connecting tubes, and a sealed port for antimicrobial dosing or sample withdrawal (Figure 1) [33]. The size of the central reservoir can vary depending on the culture volume (100–250 mL) [28].

Additionally, some one-compartment models feature filters at the outlet of the central reservoir leading to the waste reservoir, preventing the loss of microorganisms during waste removal to mimic antimicrobial elimination. However, a major limitation of these models is the potential clogging of the membrane filter, particularly when simulating antimicrobials with short half-lives, due to the high flow rate [34,35]. Ensuring continuous homogeneity of the culture in the central reservoir by using magnetic stirrer bars may help to mitigate this to some extent, particularly when flow rates are relatively low when working with drugs with relatively long half-lives [36].

For measurement of the outcome, i.e., the experimental pharmacodynamic end-point, bacterial samples are collected from the central reservoir to determine the concentration of the bacteria over the course of treatment with a simulated clinical dosing regimen. Traditional quantitative cultures are then used to determine the concentration of bacteria in the samples, expressed as the number of colony-forming units per millilitre of sample (CFU/mL). The data generated will describe the time course of change in the bacterial population over the duration of treatment. The data analysis can be either descriptive or more advanced using mechanism-based PK/PD modelling [37,38].

The one-compartment model has been extensively used in the preclinical evaluation of antimicrobial regimens either in combination or as monotherapy. For, instance, it was used to demonstrate the microbiological efficacy of the synergistic combination of cefazoline with vancomycin or daptomycin against daptomycin- and vancomycin-resistant *Staphylococcus aureus* [39]. In another example, it was used to demonstrate the activity of standard dosing regimens of ceftazidime/avibactam against Klebsiella pneumoniae carbapenemase (KPC)-producing strains [40].

In general, a one-compartment *in vitro* infection model serves as a robust tool for evaluating the PK/PD of antimicrobials [41,42]. It is also relatively inexpensive compared with other models and has thus been extensively used to study the effects of both monotherapies and combination antimicrobial therapies [43,44]. Nevertheless, the model does have its limitations, including the potential dilution of microorganisms, loss of microorganisms in models lacking filters, and high requirements for growth media and antimicrobials [45]. To minimise the impact of loss of organisms in the pharmacodynamic analysis of kill curves, a mathematical model can be applied to make accurate corrections [46]. However, a one-compartment model without filters may not be suitable for quantifying the emergence of resistance due to the loss of bacteria during the experiment [33].

### 2.2. Hollow Fibre Infection Model

The HFIM is a compartmentalised infection model designed to expose microorganisms to varying antimicrobial concentrations, mimicking the concentration fluctuations in human tissue or plasma [14]. The model setup comprises different components, including a hollow fibre cartridge, the central reservoir, diluent reservoir, waste reservoir, pumps, and connecting circuits (Figure 3A) [47].

The cartridge is a compact, single-use component of the HFIM and comes with different pore sizes. It consists of thousands of hollow fibres arranged in a parallel array along the axial direction of its cylindrical structure [48]. Its high surface area-to-volume ratio and small volume of extracapillary space (ECS) (10–20 mL) (Figure 3B) enables rapid antimicrobial distribution [17,47,49] within the central compartment of the HFIM. The fibres vary in their percent of porosity, molecular weight, hydrophobicity, and permeability. There is a variation in the permeability of the fibres depending on pore size, membrane surface area, length, arrangement of the fibres, and fibre type [47].

Overall, the cartridge has two spaces separated by a semi-selective membrane bundle of microfibres. The fibres’ walls act as a selective membrane filter, dividing the lumen of the fibres (intracapillary space) from the ECS, which is the area between the outer surface of the fibres and the sealed shell of the cartridge [17,47].

#### 2.2.1. Procedures of Hollow Fibre Infection Model

The microorganisms are introduced into the ECS of the cartridge via the sampling port (Figure 3A) [50]. The semi-selective pores of the fibres enable the exchange of gases, nutrients, metabolic wastes, and antimicrobials while retaining the infecting organisms in the ECS [51]. By retaining microorganisms in the ECS, their loss and dilution during pharmacokinetic decay are avoided, unlike in the one-compartment model. The antimicrobial is usually administered to the central reservoir with a syringe pump in a dosage mimicking its intended regimen for clinical use [52]. The drug solution in the central reservoir is pumped to the cartridge with a pump (e.g., duet pump) at a high flow rate (80–120 mL/min) to ensure the rapid distribution of the antimicrobials and nutrients into its ECS containing the microorganism [32]. The fluid in the cartridge is pumped back to the central reservoir. Similarly, the growth media from the diluent reservoir is pumped into the central reservoir at a rate equivalent of the clearance of the drug, mimicking the half-life of the antimicrobials. Simultaneously, drug-containing growth media from the central reservoir is pumped out to the waste reservoir [53]. The equivalent rate of renewal and removal of growth media from the central reservoir keeps a constant volume of distribution in the central compartment. Similar to the one-compartment model and all dynamic PK/PD models per se, calibration of the pumps and adjustment of the flow rate of media based on the desired pharmacokinetic profile of the antimicrobial is critical to ensure appropriate antimicrobial exposures are evaluated by the model. Fluid samples can be taken from the central compartment or the ECS to confirm the simulated pharmacokinetic profile of the test antimicrobials [14].

For end-point measurement, microbial samples are taken via the sampling ports to characterise microbial density and emergence of antimicrobial resistance during the study period. The time-course of change in the total bacterial concentration is determined by quantitative cultures using drug-free agar plates. The emergence of any resistant sub-populations is quantified using agar plates impregnated with the antimicrobial of interest. The change in bacterial density (CFU/mL) over time can be subject to mechanism-based mathematical modelling for robust analysis [54,55]. In addition, susceptibility testing can also be performed from any growth observed during treatment to determine the shift in MIC following exposure to treatment [56]. Samples of any resistant sub-populations can be analysed by whole genome sequencing (WGS) to describe the mechanisms of resistance [57].

#### 2.2.2. The Importance and Limitations of the HFIM

The HFIM is an invaluable tool and is often considered the “gold standard” *in vitro* PK/PD infection model for the evaluation of treatment regimens that aim to suppress the emergence of resistance; it has been increasingly used in the development of antimicrobial regimens for various difficult-to-treat infections. For instance, tedizolid, ceftriaxone-avibactam, ertapenem-clavulanate, vancomycin, linezolid, and moxifloxacin have been evaluated in HFIM to develop alternative anti-tuberculosis treatment regimens [58,59,60,61,62].

Unlike one-compartment systems, there is no dilution or loss of microorganisms in HFIM, as they are kept in the ECS of the cartridge [51]. Therefore, HFIM is an ideal tool for studying the emergence of microbial resistance, unlike the one-compartment model. HFIM allows repeated sampling to increase the reproducibility of conducting long-term experiments without the risk of contamination [32]. In addition, the natural growth kinetics of microorganisms in the human body can be represented in HFIM, as it provides a high surface area for the growth [32,47]. Furthermore, HFIM avoids the ethical issues required for conducting *in vivo* studies [14,63].

However, HFIM has some limitations. First, the concentration of some antimicrobials in the central compartment may drop because of drug binding to the plastic components [14,15] and enzymatic degradation linked to the accumulation of the bacterial enzymes in the ECS of the cartridge [64]. Second, although it allows evaluation of antimicrobial activity unconfounded by the immune system and is perhaps representative of immunosuppressed patients, HFIM does not account for the antimicrobial actions of the immune system of the host, which is important in determining the overall outcome of the antimicrobial therapy [18]. Thirdly, traditional HFIM studies do not simulate exposure of microorganisms to the potential wide temperature and pH ranges that patients experience, although this is technically possible albeit challenging [18,47]. Lastly, HFIM is expensive compared with other *in vitro* methods [15,28], which is mostly associated with the single use of the cartridges, the high volume of media, particularly for short half-life drugs, and the labour-intensive procedures in conducting the experiments.

### 2.3. Dynamic In Vitro Biofilm Model

Microorganisms can grow independently as free-floating cells (planktonic) or within specialised multicellular structures called biofilms. A biofilm consists of microorganisms, particularly bacteria, that form a complex architecture of living cells and extracellular matrix. This structure prevents immune cells and antimicrobials from accessing the bacteria. Biofilms enable organisms to persist long-term and develop resistance mechanisms against antimicrobials, especially in cases of chronic infections [65,66]. The treatment of such chronic infections needs to consider antimicrobials that can disrupt the biofilm-forming bacteria.

There are several methods used for evaluating the antibiofilm activity of antimicrobials, including the microtiter plate, glass beads, sorbarod model, CDC reactor, drip-flow methods, and continuous-flow model [67,68,69,70]. However, most of these models do not simulate the time course change in the concentration of antimicrobials in the human body, except the CDC reactor [71] and sorbarod *in vitro* models [72].

#### 2.3.1. Sorbarod *In Vitro* Biofilm Model

Sorbarod is a compact cellulose fibre structure with filters at the bottom of its cylindrical structure. The model is inoculated with bacteria with a sterile syringe. The inoculum is incubated at 37 °C for 24 h. After incubation, the cells adhered to the cellulose fibre will be perfused continuously with a sterile broth until a stable biofilm mass is formed or until a constant colony of bacteria is eluted [72,73,74]. The bacterial population eluted characterises the biofilm-embedded cells on the surface of the cellulose fibre.

The antimicrobial is administered via infusion into the central reservoir (Figure 4), which is diluted with a constant flow of drug-free growth media from the diluent reservoir, simulating the time course antimicrobial exposure in patients with infections. The mixture of the antimicrobial and the growth medium is perfused into the biofilm-containing sorbarod by a peristaltic pump, exposing the biofilm to changing drug concentrations. The antibiofilm activity is determined by enumerating the number of colonies of bacteria harvested from the sorbarod by vortexing or sonicating within a growth medium at the end of the experiment [72,73,74].

The sorbarod dynamic *in vitro* model has been used for evaluating the antibiofilm efficacy of different antimicrobials, including, for example, beta-lactams and fluoroquinolones against *P. aeruginosa* and *Streptococcus pneumoniae* [75,76].

While the sorbarod model has applications for evaluating the antibiofilm activity of the antimicrobials, it is not suitable when strict uniformity of conditions is required for reliable or reproducible results due to its small and cylindrical physical scale and irregular structure of cellular fibres [67]. Additionally, unlike the CDC reactor model, the sorbarod method does not allow for repeated investigation of the biofilm from the same device; the biofilm can be examined only at the conclusion of the experiment. To enable multiple analyses of the bacterial biofilm, multiple sorbarod devices must be prepared for the same bacterial isolate [67,77].

#### 2.3.2. CDC Biofilm Infection Model

The CDC reactor model consists of a one-litre beaker having a side effluent arm for the elimination of media fluids from the central reservoir to the waste reservoir (Figure 5) [78,79]. The lid of the reactor has three ports for the administration of bacterial growth media, fluid sample collection, and air ventilation. Additionally, the lid holds eight suspended rods, each with three housings for coupons, providing surfaces for biofilm-forming bacteria. This setup accommodates a total of 24 sampling coupons. The pharmacodynamic activity of antimicrobials is assessed by sampling the biofilm formed on the coupon surfaces. The beaker’s contents, including antimicrobials and growth media, are thoroughly mixed by a stir bar or vane driven by a magnetic stir plate, which generates high shear [71,80].

The PK/PD biofilm model experiment has three phases: batch phase, continuous phase, and PK/PD phase [33].

The batch phase of the biofilm model allows the initial bacterial growth and the attachment of biofilm-forming bacteria on the surface of the coupons in the reactor. Before starting the batch phase, a single colony of bacteria from an overnight incubated agar plate is inoculated into Tryptic Soy Broth (TSB, 100 mL) and incubated for 24 h in a shaking incubator. To initiate the batch phase, first, the reactor is placed on stirrer hot plate or on a magnetic stirrer in an incubator. The inoculum injection port is flushed with TSB using a luer syringe to ensure that inoculum reaches the batch media (500 mL of TSB). Then, 1 millilitre of the 100 mL overnight culture bacterial inoculum is injected into the reactor containing TSB to achieve an inoculum concentration ~1 × 10^8^ CFU/mL. The stirring bar of the reactor is rotated at 125 RPM for 24 h to initiate bacterial attachment on the surface of the coupons. It is important to control the rotation rate of the stirrer bar to ensure an optimal shear stress to allow the attachment of bacteria on the surface of the coupons and maintain a good density of the viable biofilm [33,81].

The continuous phase is initiated by the aseptic assembly of the necessary connecting tubes for continuous media flow and waste removal. Determination of the flow rate of growth media appropriate for the study organism is crucial before starting the continuous phase of the experiment. The flow rate should ensure the residence time of the media in the reactor is shorter than the doubling time of the bacteria so that the non-adhered bacteria are removed and only biofilm-attached cells remain in the reactor [33]. This can be estimated based on the ratio of the reactor volume and the organism’s residence time (e.g., 350 mL/30 min = 11.67 mL/min for *Pseudomonas aeruginosa*). At this specific rate, TSB is run continuously for 24 h to promote the development of the biofilm [33,80,82].

In the therapeutic phase, Cation-Adjusted Muller Hinton Broth (CAMHB) is first introduced to the model to provide sufficient nutrients for the bacterial growth. Then, the media flow rate is adjusted to a rate mimicking the clearance of the test antimicrobial [33]. The therapeutic phase begins with the injection of the antimicrobial solution into the CDC bioreactor while keeping the entire setup at 37 °C for the duration of treatment. Sample coupons are taken at specific time intervals to assess the therapeutic response of the biofilm-embedded bacteria [82,83].

The sampled coupon is first rinsed in normal saline to eliminate planktonic bacteria and antimicrobials. Next, biofilm-embedded bacteria are removed from the coupon’s surface by vortex mixing and sonication in normal saline. The resulting bacterial suspension is then serially diluted in test tubes [84]. The diluted suspension is plated on agar medium and incubated at 37 °C for 24 h. Finally, the antibiofilm efficacy of the antimicrobials is expressed in CFU/cm^2^ surface area of the coupons and compared with the growth control arms [85,86].

In addition, other sample coupons are taken to characterise the biofilm architecture with a microscope. After removing the planktonic cells in normal saline, the biofilm at the surface of the coupons is fixed with reagent kits and dried [87]. The biofilm structure is then assessed using a scanning electron or confocal laser microscope [80,83,85].

The major differences between the CDC reactor and the sorbarod model lie in two key aspects. First, the CDC reactor allows for the study of the time-course antibiofilm effects of antimicrobials by enabling the sampling of coupons at multiple time points throughout the experiment. In contrast, the sorbarod model permits sampling only at the end of the experiment, which requires dismantling the setup. Second, the structure of the biofilm in the sorbarod model may be heterogeneous due to the physical scale and irregularity of the cellulose fibre, whereas the CDC reactor typically provides more uniform conditions for biofilm formation [67,77,88].

The CDC biofilm infection model has been used to evaluate treatment regimens against pre-formed biofilms of common biofilm-forming bacteria, including *P. aeruginosa* and *S. aureus.* For instance, the model was used to demonstrate effective bacterial killing by the combination of colistin with ceftolozane/tazobactam or meropenem against biofilm-embedded, extensively drug-resistant (XDR) and multi-drug-resistant (MDR) *P. aeruginosa*, respectively [83]. In other examples, the model was used to demonstrate the successful eradication of biofilm-embedded *S. aureus* strains by the combination of daptomycin or linezolid with fusidic acid [89], levofloxacin and rifampicin [90], or dalbavancin and rifampicin [91].

In simulating clinical treatment, an important challenge for PK/PD biofilm models, including those using the CDC reactor, is the lack of data on the exposure (concentration) of antimicrobials at the site of biofilm infection, particularly for some infections such as those associated with orthopaedic implants. Simulated profiles are often extrapolated from plasma pharmacokinetics and therefore may have limitations in the translation of findings. Another limitation is that, like other *in vitro* models, the CDC reactor model does not account for the role of the immune system of the host [68].

### 2.4. Dynamic In Vitro Bladder Infection Model

When the bladder infection is treated, the infecting microorganisms are exposed to varying concentrations of antimicrobials throughout the treatment period due to the difference in pace of urine formation and voiding [92]. In addition, the concentration of the antimicrobial in the bladder varies with the types of antimicrobial administered based on their renal clearance [93]. The dynamic *in vitro* bladder infection model was developed to simulate the humanised pharmacokinetics of the antimicrobials and the bacterial growth kinetics in the bladder.

#### Procedure of Dynamic *In Vitro* Bladder Infection Model

The dynamic in vitro bladder infection model is an automated system designed to control the rate of urine formation and micturition over an extended period, simulating the urodynamic features of the bladder [94,95]. The mimicry of human urodynamics is achieved through the use of peristaltic pumps integrated into the model [92]. As depicted in Figure 6, the model includes various reservoirs representing the intestinal compartment, the circulatory compartment, and the bladder, all interconnected by tubes [94,96].

To initiate the treatment, the reservoir simulating the bladder is infected with standardised bacterial concentration of the target inoculum [92,97,98]. The growth medium, such as pooled urine, synthetic human urine, or standard nutrient broth, is added in a media reservoir [93,97]. The media is pumped by a peristaltic pump into the intestinal absorption compartment, where the antimicrobial is injected and mixed with a magnetic stirrer. The antimicrobial solution is then pumped to the central “circulatory” compartment to simulate intestinal absorption. The fluid pumping rate into the intestinal and circulatory systems is kept equivalent to ensure a static fluid volume in these compartments [92,96,97]. The fluid in the circulatory system is distributed into the bladder-simulating reservoir; multiple bladder reservoirs could be connected with a cumulative flow rate matching the flow into the circulatory compartment [96]. The volume of the bladder compartment is monitored to gradually increase the fluid volume over time, with intermittent voiding of waste into the waste reservoir to simulate the human micturition physiology [95]. The fluid voiding rate is typically set to every four hours, leaving a residual fluid in the bladder compartment [94,95,98]. The variation in the rate of fluid flow and the fluid volume in the bladder compartment simulates the dynamic exposure of bacteria to antimicrobials.

Samples of bacteria and antimicrobials are taken from the bladder compartment at regular intervals throughout the study period. The bacterial killing activity is then assessed by plating the serially diluted bacterial suspensions on appropriate agar plates [99]. This model can also be used to study the emergence of antimicrobial resistance [98,99].

The bladder infection model has been used to evaluate antimicrobials commonly used for the treatment of urinary tract infections. Typical examples include the works of Abbott and colleagues [92,96,100]. For instance, they have used this model for a robust pharmacodynamic profiling of oral fosfomycin against *Enterococcus faecalis* and *Enterococcus faecium,* including the identification of urinary concentration-based PK/PD targets for the optimisation of fosfomycin regimens [92]. Similarly, Abbott et al. have used this model to derive urinary PK/PD targets for ciprofloxacin and predict optimal dosing regimens for oral administration based on the susceptibility profile of the pathogen investigated (*E. coli*) [96].

The bladder infection model is not without limitations. Although the accumulation and voiding of urine in the bladder is simulated to reflect the physiologic micturition process, it does not perfectly simulate the natural process controlled by the delicate anatomical features of the bladder, potentially affecting the *in vivo* representation of the exposure–response relationship studied in the *in vitro* systems [92,101]. Another important limitation is the variation in the nutritional components of the media (simulated urine) used in the model, which may also affect the translation of exposure–response relationships observed in the *in vitro* system [92].

### 2.5. Dynamic In Vitro Aspergillus Infection Model

The dynamic *in vitro* aspergillus infection model was developed to bridge the translation of *in vitro* pharmacodynamic and pharmacokinetic findings to clinical settings. The pathogenesis of *Aspergillus fumigatus* in the respiratory tract is experimentally modelled by an *in vitro* bilayer cell culture of endothelial and alveoli cells [102]. This in vitro infection is then exposed to dynamic concentrations of antifungal agents and subsequently evaluated for therapeutic response using a biomarker [103,104].

#### Procedures of Dynamic *In Vitro* Aspergillus Infection Model

The schematic of the fluid circulation of the model is presented in Figure 7. The *in vitro* bilayer cell culture of the endothelial and alveoli cells in a bioreactor is prepared using cell culture well inserts [102,103,105]. The alveolar surface of the bilayer cell culture is then inoculated with the fungal suspension of 1–3 × 10^4^ conidia/ml [103]. The cell culture insert is housed in a stainless steel bioreactor [103,105] and incubated at 37 °C [103,104]. The assembled bioreactor is connected to the central reservoir containing the antifungal agent through a connecting tube (Figure 7) [103,104].

The drug in the central compartment is diluted with growth media (endothelium basal medium-2 (EBM-2) supplemented with foetal bovine serum (FBS)) [103,104,105]. This mixture of fungal growth media and the antifungal diffuses into the infected endothelial cells, mimicking the exposure of aspergillus species to the antifungal agent in the human alveoli. The mixture of the antifungal agent and the fungal growth medium is pumped by a peristaltic pump at regular time intervals, depending on the pharmacokinetics of the antifungal agent, into the bioreactor [106].

The simulated pharmacokinetics of the antifungals are confirmed by measuring the antifungal concentration using chromatographic techniques, whereas their pharmacodynamic activities are assessed by measuring the level of galactomannan released from the fungus [107,108]. The level of galactomannan released is used as a surrogate measure of the viable fungal cells, and hence, the antifungal activity of the agents being investigated; a progressive decrease in the level of this biomarker correlates with fungicidal activity [102].

A typical example of the application of the aspergillus infection model in the development and optimisation of antifungal agents is the work of Negri et al., which characterised the pharmacodynamics of a novel antifungal agent F901318 for acute sinopulmonary aspergillosis caused by *Aspergillus flavus* [105]. Other examples include studies on voriconazole and isavuconazole [103,104,106].

An important limitation of this model is that it mimics only the early, invasive phase of the fungal infection. In addition, like most other *in vitro* infection models, it does not emulate the contribution of the immune system to the activity of the antifungal agents [104].

Table 1 summarises the various PK/PD *in vitro* infection models discussed in this paper, outlining their utility, advantages, and limitations.

## 3. Simulating the Pharmacokinetics of Antimicrobials

Dynamic *in vitro* infection models are used to describe the time course of response to antimicrobial therapy when microorganisms are exposed to the dynamic free concentrations of the antimicrobials as they occur in the human body [99,112]. The pharmacokinetics (dynamic concentration–time profile) of antimicrobials in *in vitro* infection systems is simulated using methods described by Blaser et al. [113] and Kesisoglou et al. [114]. This involves controlling the flow of drug-free media into and out of the central reservoir to mimic the elimination pharmacokinetics of antimicrobials in the human body [115]. The models can be used to simulate the pharmacokinetics of both combination and monotherapy regimens. For monotherapy simulation, the fluid flow rates to and from the central compartment are adjusted based on the clearance of the antimicrobial being tested. One-compartment kinetics is usually simulated by calculating the “humanised clearance” of the study drug that corresponds to the volume of the distribution of the *in vitro* system (which can be specified as needed, usually ranging from 100 to 500 mL) to keep the same half-life of elimination as in humans (Equation (1)).
(1)$$Cl=\frac{0.693\times V_{C}}{t_{1/2}}$$
where *Cl* is the humanised clearance of the antibiotic, *t*_1/2_ is the half-life, and *V_C_* is the volume of the central compartment.

When simulating the pharmacokinetics of the combination of antimicrobials with a distinct clearance or half-life, modification of the dynamic *in vitro* setup (Figure 8) is needed. The setup can be configured in either a serial or parallel manner to accommodate the distinct pharmacokinetic profiles of the drugs [114].

The first serial setup in a dynamic infection model for two antimicrobials that differ in half-life was developed by Blaser in 1985 [113]. In this model, the net inflow of media into the central reservoir is set based on the clearance of the antimicrobial with a shorter half-life and higher clearance rate. For instance, to simulate the pharmacokinetics of a combination of two antimicrobials, A and B, in HFIM, where the clearance of A is greater than that of B, the flow rate of the *in vitro* system is adjusted according to the clearance of antimicrobial A [113]. Since antimicrobial B is eliminated more rapidly than expected, it is supplemented into the central reservoir at a concentration equivalent to its target concentration. This is achieved by pumping antimicrobial B from the supplementary reservoir to the central compartment at a flow rate equivalent to the difference in clearance of the two antimicrobials, offsetting its excess elimination associated with the rapid clearance of the *in vitro* system [113]. To determine the dose of antimicrobial B for supplementation, the volume of its supplementary reservoir (*V_B_*) is first determined using Equation (2) [113].
(2)$$V_{B}=\frac{{Cl}_{A}-{Cl}_{B}}{{Cl}_{B}}V_{C}$$
where *Cl_A_* and *Cl_B_* are the clearances of antimicrobial A and B, respectively, and *V_C_* is the volume of the central compartment.

The bolus mass of antimicrobial B (*M_B_*) injected into the central compartment is determined using Equation (3).
(3)$$M_{B}=C_{B}max\cdot Vc$$
where *C_B_max* and *V_C_* are the peak concentration of B and the volume of fluid, respectively, in the central compartment.

The infusion rates of antimicrobial B into the serial vessel or supplementary reservoir (*I_B,S_*) are determined based on the infusion rate to the central compartment (*I_B,C_*) and the volumes of the fluids in the central compartment and the supplementary reservoir (Equation (4)).
(4)$$I_{B,S}=\frac{V_{B}}{V_{C}}I_{B,C}$$

The amount of antimicrobial B (*m_B_*) bolus injection into the supplementary reservoir within a short period is calculated as
(5)$$m_{B}=\frac{V_{B}}{V_{C}}M_{B}$$
where *M_B_* is the mass of antimicrobial B bolus injected into the central vessel, and *V_B_* and *V_C_* are the volumes of fluid in vessel B and in the central compartment, respectively.

Pharmacokinetic profiles for combinations of three or more antimicrobials, each with differing clearance or half-life, could be simulated using a serial or parallel configuration setup, as described by Kesisoglou et al. [114]. Implementing the serial arrangement of the *in vitro* system involves determining the antimicrobial flow rate and the volumes of broth in each antimicrobial-containing vessel. In a serial configuration of HFIM for three antimicrobials (Figure 9), for instance, the antimicrobial with the highest clearance (antimicrobial A) will be bolus dosed and/or infused into the central compartment. The other antimicrobials (antimicrobials B and C) are administered (bolus or infusion) into both the central compartment and their respective serial vessels.

In serial configurations, unlike the Blaser *in vitro* design from 1985 [113] for two-antimicrobial combinations, Kesisoglou et al. [114] made the following modifications for the combination of three or more antimicrobials.

The volume of the serial vessel or feeding vessel for N number of antimicrobials (A, B, C, …) has the flexibility of being changed if the elimination rate constant of antimicrobial A (*K_A_*) is greater than the elimination rate constant of the other combinatory antimicrobials (*K_B_*, *K_C_*, etc.) and *V_A_K_A_* > *V_B_K_B_* + *V_C_K_C_* + …, where *K_A_*, *K_B_*, and *K_C_* and *V_A_*, *V_B_*, and *V_C_* are the elimination rate constants and the respective volumes of the fluid containing antimicrobials A, B, and C, respectively.The injection period for each antimicrobial in all vessels should not necessarily be large.The series design does not necessarily require that the maximum concentration of each antimicrobial solution flowing to the central compartment from its respective supplementary reservoir is equal to its concentration in the central compartment; i.e., *C_i_^in max^* ≠ *C_i_^max^* for any antimicrobial i because there is a flexibility in changing the ratios of the vessel fluid volumes, where *C_i_^in max^* is the maximum concentration of the antimicrobial i flowing to the central compartment, and *C_i_^max^* is the maximum concentration of the same antimicrobial in the central compartment shortly after administration.

Representations of the symbols for dosing, concentrations, volumes, and the flow rates of three or more antimicrobials (A, B, C, …) in the serial HFIM of Figure 9 are presented in Table 2.

In the serial configuration, the dose of antimicrobial i injected into the central compartment (*M_i_*) and each supplementary reservoir (*m_i_*) is determined using Equations (6) and (7).
(6)$$M_{i}={(Ci}^{max}-C_{i}(0))V_{c}\;\;for\;i=A,B,C,\;\ldots$$
(7)$$m_{i}={Ci}^{max}V_{C}\left(\frac{t_{\frac{1i}{2}}}{t_{\frac{1A}{2}}}-1\right)⟹\;m_{i}={Ci}^{max}V_{C}\left(\frac{K_{A}}{K_{i}}-1\right)\;\;for\;i=B,C,\;\ldots$$

The potential of the serial HFIM design to predict the pharmacokinetics of the combinations of levofloxacin, meropenem, and ceftazidime was experimentally validated by Kesisoglou et al. [114]. This study revealed that the half-life and the target concentration of the studied antimicrobials were within the acceptable range of their target pharmacokinetic parameters with slight deviations [114]. This suggests that the series HFIM design holds promise as a tool for predicting the pharmacokinetics of antimicrobial combinations *in vitro*.

In the parallel configuration of HFIM (Figure 10), the supplementary reservoir of each antimicrobial is arranged in such a way that it can feed its contents to the central reservoir. This HFIM setup involves dosing each antimicrobial i into the central compartment and the corresponding reservoir [114]. Each antimicrobial flows into the central compartment at a rate simulating the clearance of the individual antimicrobial in the human body.

In the parallel configuration of the HFIM, the bolus dose of each antimicrobial i into the central compartment is determined by Equation (8):(8)$$M_{i}={(Ci}^{max}-C_{i}(0))V_{c}\;\;for\;i=A,B,C,\;\ldots$$
where *Ci^max^* is the peak target concentration of the antimicrobial i, *C_i_*(0) is the concentration of the antimicrobial i at time 0 if any, and *V_C_* is the total volume of the central compartment.

The amount of antimicrobial i (*m_i_*) injected into the corresponding vessel is determined by Equation (9):(9)$$m_{i}={Ci}^{max}\left(\frac{F^{out}}{K_{i}}-V_{c}\right)\;\;for\;i=B,C,\;\ldots$$
where *C_i_^max^* is the peak concentration of antimicrobial i in the central compartment, *F^out^* is the fluid flow rate from the central compartment to the waste reservoir, *K_i_* is the fluid flow rate constant of antimicrobial i, and *V_C_* is the volume of fluid in the central compartment.

The parallel configuration of the HFIM was experimentally validated using the humanised pharmacokinetics of combinations of meropenem, ceftazidime, and ceftriaxone. The observed pharmacokinetic profile of the three antimicrobials was within the acceptable range of the target simulated profile [116], demonstrating that the parallel HFIM configuration is a promising design for studying the pharmacokinetics of multi-drug combination regimens *in vitro*.

## 4. Application of Dynamic *In Vitro* Infection Models in the Development and Optimisation of Antimicrobials

The application of dynamic *in vitro* infection models in antimicrobial development has been progressively increasing in the last decade. They enable basic pharmacodynamic profiling of antimicrobials to generate key initial data that could be used as a reference for further investigation in animal model experiments and clinical trials. This reduces the unnecessary waste of time and cost required for antimicrobial research and development in clinical settings [45,47]. Indeed, these models have been instrumental in the development of various antimicrobials including antibacterials [50,117,118], antifungals [105], and antivirals [119,120,121,122,123,124]. For instance, the novel antibacterials zoliflodacin [118] and lefamulin [117], aimed at treating *Neisseria gonorrhoea* infection, were initially evaluated in HFIM. Similarly, a dynamic *in vitro* aspergillus infection model was used in the development of the new antifungal, F901318, which showed promising potential for clinical application in treating fungal infections including aspergillosis [105].

A key application of dynamic PK/PD models in the initial stage of antimicrobial development is in elucidating the PK/PD exposure–response relationship of antimicrobials, thereby guiding the selection and optimisation of antimicrobial regimens. The *in vitro* system allows full control of variables for flexibility in designing the traditional dose fractional studies used to identify the PK/PD drivers of efficacy. For example, HFIM was used for dose-fractionation studies to identify free AUC/MIC ratio as the PK/PD index of gepotidacin, a novel first-in-class antibiotic, and determine the magnitude of its PK/PD index [125]. In this study, the free-drug AUC/MIC ratio needed to prevent resistance amplification was also determined over the treatment course, which would not be possible in the traditional animal model of infections used for fractionation studies. Another example is the novel beta-lactam/beta-lactamase inhibitor combination aztreonam/avibactam, for which the PK/PD drivers for both aztreonam and avibactam in the combination were derived using dose-fractionation data generated from the HFIM [126]. In addition to dose-fraction studies, advanced mathematical modelling and simulation can also be applied to data generated from dynamic *in vitro* PK/PD infection models to predict the PK/PD index of antimicrobials [127,128].

In addition to new drug development, dynamic *in vitro* infection models have been used as a key tool supporting ongoing efforts to repurpose existing antimicrobials for novel indications. Nixon et al. [129], for example, used the HFIM to evaluate the pharmacodynamics of flubendazole against *Cryptococcus neoformans* in their effort toward repurposing this antiparasitic drug for the treatment of Cryptococcal Meningoencephalitis. Another example is the work of Srivastava et al. [130], in which they used the HFIM for repurposing cefazolin, in combination with avibactam, for the treatment of drug-resistant *Mycobacterium tuberculosis* (TB).

Another key application where in which dynamic *in vitro* infection models have the most advantage is the development of alternative regimens for treating infections caused by drug-resistant microorganisms. For instance, HFIM has been widely used and recommended for the evaluation and development of alternative anti-tuberculosis combination regimens that provide synergistic rapid bactericidal activity while suppressing the emergence of resistance [131]. Such hollow fibre system tuberculosis models have been shown to have a high predictive accuracy of clinical outcomes [27]. In addition to synergistic combinations, dynamic infection models enable the optimisation of alternative modes of administration or dosing schedules that could suppress the emergence of resistance. For instance, when evaluated in HFIM, the sequential administration of sulfamethoxazole/trimethoprim, linezolid, and clindamycin resulted in suppression of the emergence of resistance in *Staphylococcus aureus* more effectively [132]. Similarly, extended and continuous infusion of aztreonam in combination with ceftazidime/avibactam showed a greater bacterial suppression effect compared with intermittent administration against *Escherichia coli* and *K. pneumonia* in HFIM [133].

## 5. Summary

The dynamic *in vitro* infection models provide reliable data on the dose exposure–response relationship of antimicrobials in a cost-effective and well-controlled manner. They provide an invaluable tool to determine the PK/PD parameters of antimicrobials to guide the dose optimisation and design of the optimal treatment schedule of antimicrobial regimens. The models optimise the effectiveness of antimicrobial regimens both in terms of microbial killing and ability to suppress the emergence of antimicrobial resistance. Hence, they play a significant role in developing new treatment options or regimens effective against antimicrobial-resistant organisms.

## Figures and Tables

**Figure 1 antibiotics-13-01201-f001:**
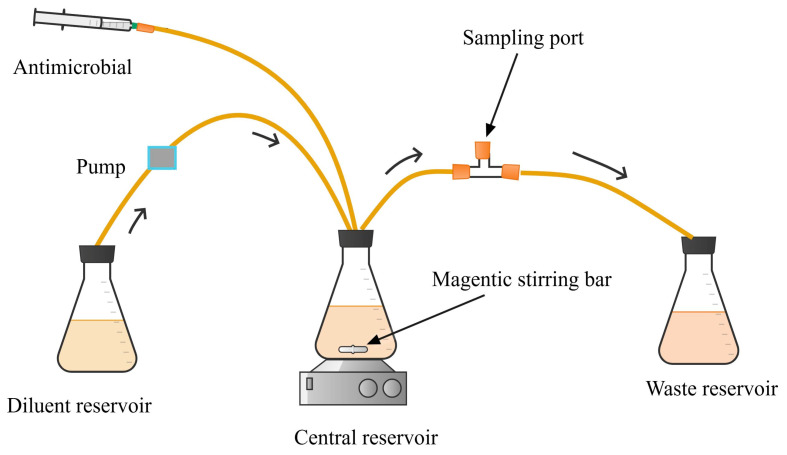
One-compartment dynamic *in vitro* infection model simulating the intravenous intermittent administration of antimicrobials.

**Figure 2 antibiotics-13-01201-f002:**
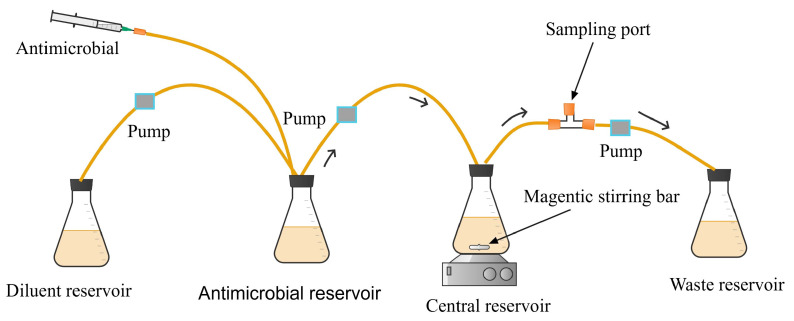
One-compartment dynamic *in vitro* infection model simulating the oral administration of antimicrobials.

**Figure 3 antibiotics-13-01201-f003:**
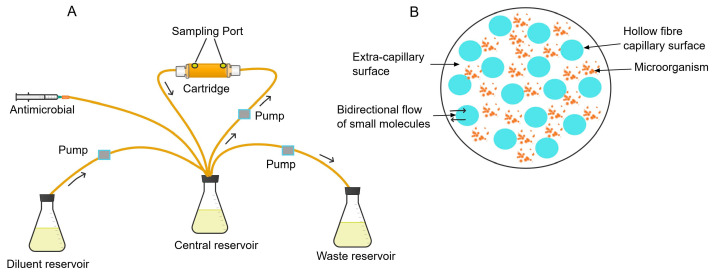
Diagram of the hollow fibre infection model (**A**) and a cross-section of the hollow fibre cartridge (**B**). The arrows “→” indicate direction of flow.

**Figure 4 antibiotics-13-01201-f004:**
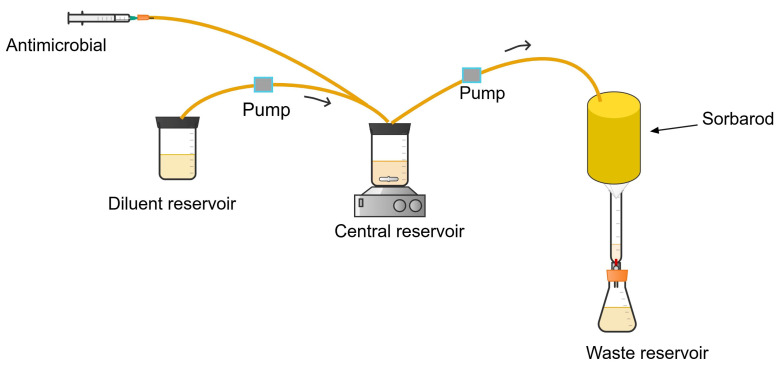
Sorbarod *in vitro* biofilm infection model.

**Figure 5 antibiotics-13-01201-f005:**
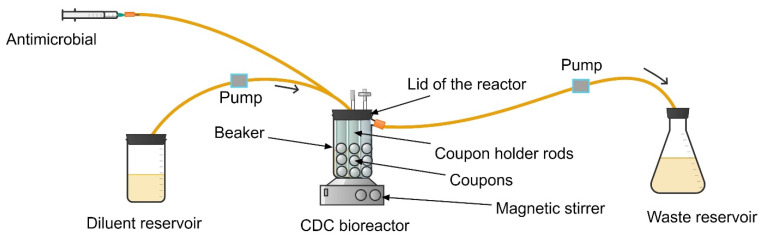
CDC biofilm reactor.

**Figure 6 antibiotics-13-01201-f006:**
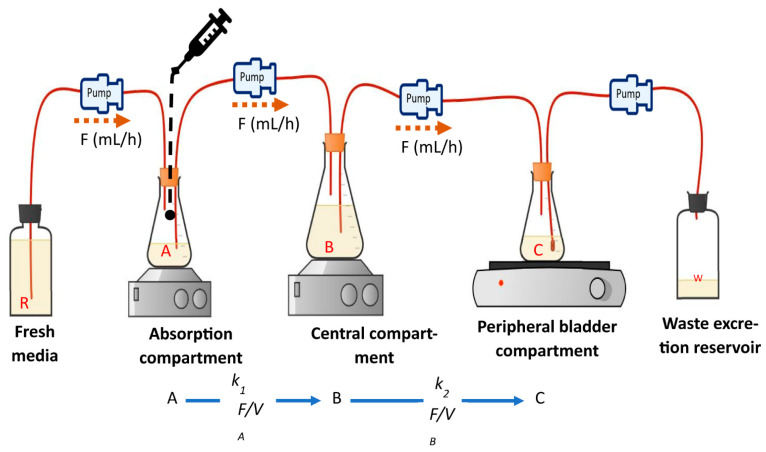
Dynamic *in vitro* bladder infection model. *K*_1_ is the absorption rate constant; *K*_2_ is the elimination rate constant to the bladder; *F* is the flow rate to the respective compartment; *V* is the respective volume of each compartment; R is the media reservoir; W is the waste.

**Figure 7 antibiotics-13-01201-f007:**
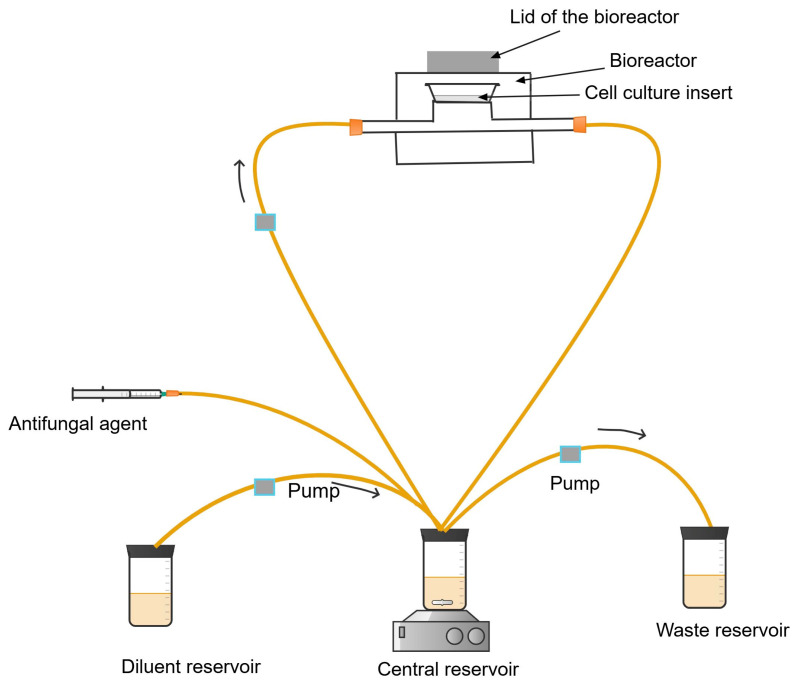
Dynamic *in vitro* aspergillus infection model consisting of the bioreactor and the circuit for the dynamic exposure of the aspergillus species to the antifungal agents.

**Figure 8 antibiotics-13-01201-f008:**
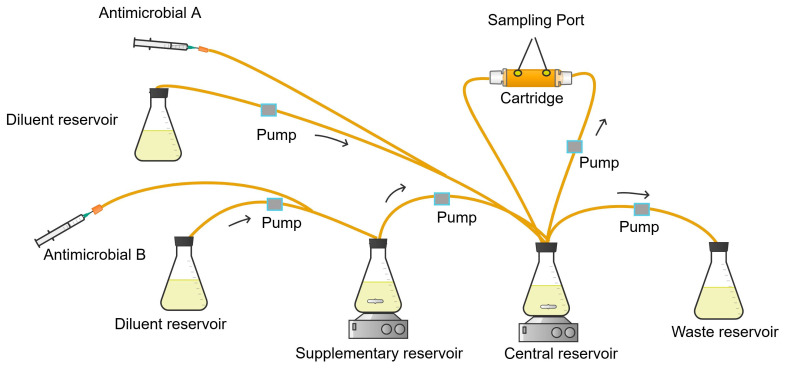
Hollow fibre infection setup for combination therapy.

**Figure 9 antibiotics-13-01201-f009:**
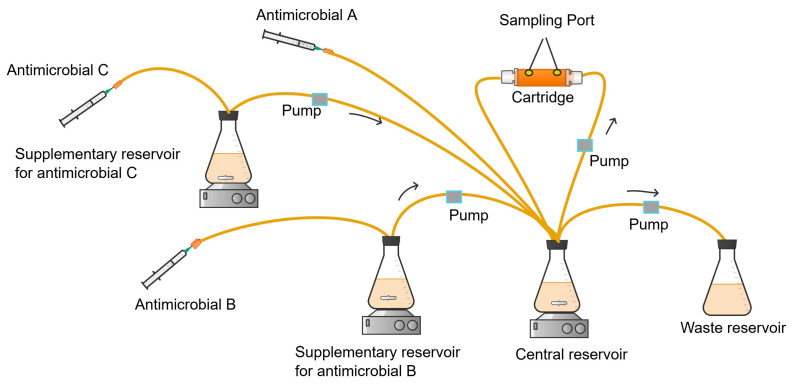
Serial configuration of HFIM for simulating the pharmacokinetics of three antimicrobials (A, B, C).

**Figure 10 antibiotics-13-01201-f010:**
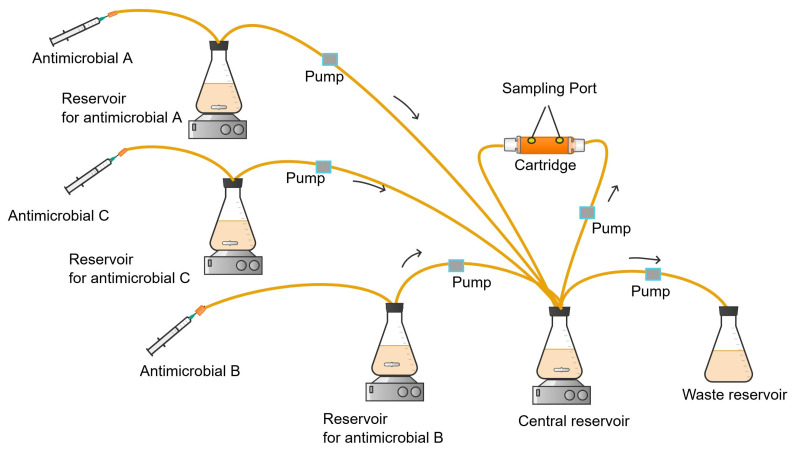
Parallel configuration of HFIM simulating the pharmacokinetics of three antimicrobials (A, B, C).

**Table 1 antibiotics-13-01201-t001:** Summary of the utility, advantages, and limitations of dynamic *in vitro* PK/PD infection models.

Model Type	Utility	Advantages	Limitations	References
One-compartment *in vitro* infection model	To study the exposure–response relationship of antimicrobials To determine PK/PD indices	Simple and easy to set up Relatively cost-effective	Dilution or loss of bacteria during the experiment High volume of contaminated waste Difficult to model short half-lives Does not account for the effect of host immune response	[17,45]
Hollow fibre infection model (HFIM)	To study the exposure–response relationship of antimicrobials To evaluate antimicrobial efficacy against drug-resistant organisms To determine PK/PD indices	Large surface–volume ratio conducive for microbial growth No loss or dilution of microorganisms A closed system with minimal biosafety risks Suitable for studying a range of pathogens, including bacteria, fungi, and viruses. Useful to study PK/PD of antibiotics used in long-term infection; example, tuberculosis	High cost Some drug may bind to the HF cartridge Does not account for the effect of host immune response	[14,17,27,45]
CDC biofilms infection model	To evaluate the antibiofilm activity of antimicrobials over time To study the biofilm-related antibiotic resistance	Allows sampling at multiple time points Enables testing of biofilms formed on surfaces of various materials (custom-made coupons) Enables simultaneous assessment of activity against planktonic bacteria	High shear forces may limit the biofilm biomass Difficult to mimic the concentration of antimicrobials at infection sites (for example, at the surface of implants) The development of biofilms may take time Does not account for the effect of host immune response	[68,81,88]
Sorbarod biofilm model	To assess the long-term antibiofilm effects of antimicrobials To study the relationship between exposure and antibiofilm efficacy of antimicrobials	Easy to set up Cost-effective model Allows sampling of planktonic cells Mimics the slow development of the *in vivo* biofilm under low shear stress conditions	Does not allow multiple sampling from a device over time Heterogenous biofilm structure Does not account for the effect of host immune response	[67,88,109]
Bladder infection model	To evaluate the efficacy of antimicrobials used for treating urinary tract infection To describe urinary specific PK/PD indices of antimicrobials	Flexible to customise bladder environment Multiple bladder setups can be run simultaneously	May not account for variations in patients’ urine composition, hydration status, or underlying health conditions over time Does not fully replicate the physiologic micturition process Limited to study the efficacy of antimicrobials used for the treatment of urinary tract infection Does not account for the effect of host immune response	[92,101,110]
Aspergillus infection model	To study the exposure–response relationship of antimicrobials against *aspergillus* spp.	Mimics the cellular environment of the host infection site	Mimic only the early invasive infection May be technically challenging to set up Bioreactors are custom made, not widely available commercially Does not account for the effect of host immune response	[104,111]

**Table 2 antibiotics-13-01201-t002:** The symbols for antimicrobial dosing amount, concentration, flow rates of fluids, and the corresponding volumes of the vessels in serial and parallel configurations of HFIM for two or more antimicrobials (A, B, C, …).

Symbol	Description	Remark
*F_i_*	Flow rate of fluid containing antimicrobial i into and out of the corresponding supplementary reservoir	*F_i_* = *F_A_*, *F_B_*, *F_C_*, …
*Ki*	Flow rate constant or elimination rate constant per unit time of fluid containing antimicrobial i	*Ki* = *K_A_*, *K_B_*, *K_C_*, …
*F^out^*	Total fluid flow rate from the central vessel to waste	*F^out^* = *F_A_* + *F_B_* + *F_C_*…
*V_i_*	Liquid volume in supplementary reservoir for antimicrobial i	*V_i_* = *V_B_*, *V_C_*, …
*V_C_*	Liquid volume in central compartment	
*m_i_*	Mass of antimicrobial i in bolus injection to supplementary reservoir i	i = B, C, …
*M_i_*	Mass of antimicrobial i in bolus injection to central compartment	i = A, B, C, …
*t_1/2i_*	Half-life of antimicrobial i	i = A, B, C, …
*C_i_^max^*	Maximum concentration of antimicrobial i in the central compartment	i = A, B, C, …
*C_i_(0)*	Initial concentration of antimicrobial i in the central compartment	i = A, B, C, …

## Data Availability

No new data were created or analyzed in this study.

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
