# Peer review of "Dynamic *In Vitro* PK/PD Infection Models for the Development and Optimisation of Antimicrobial Regimens: A Narrative Review"

_antibiotics, 2024, doi:10.3390/antibiotics13121201_

Round 1
Reviewer 1 Report
Comments and Suggestions for Authors
Manuscript title: Dynamic in vitro infection models for the development and optimisation of antimicrobial regimens: a narrative review
Summary: This review summarizes various dynamic in vitro infection models, including one-compartment, hollow fiber, biofilm, and specialized infection models, which simulate human-like antimicrobial concentration-time profiles to enable reliable pharmacokinetic/pharmacodynamic studies. These models provide valuable data for defining initial dosing in clinical trials and optimizing treatments against resistant pathogens.
Comments:
The manuscript is well-written and provides detailed information about various in vitro dynamic infection models. However, the manuscript could be improved by considering the following comments:
1. Rationale for Model Selection: There are several other dynamic infection models available, such as Chemostat and Turbidostat Models, Microfluidic and Organ-on-a-Chip Models, Perfusion Bioreactor Models, Dialysis Tubing Models, and Three-Dimensional (3D) Spheroid and Tissue Models. Please provide a rationale for selecting only the one-compartment dynamic in vitro model, Hollow Fiber Infection Model (HFIM), dynamic in vitro biofilm model, dynamic in vitro bladder infection model, and dynamic in vitro aspergillus infection models. Explain the specific reasons these models were highlighted.
2. Detailed Discussion for Each Model: While sufficient details are provided for each model, consider expanding the discussion to include:
a. Situational Applicability: Describe the scenarios where each model is most appropriate and any circumstances under which a model may be unsuitable.
b. Real-World Applications: Provide examples of real-world studies or cases where these models have been applied to generate impactful data.
c. Endpoints and Data Analysis: Include information on the key endpoints measured for each model, the types of data generated, and the methods used to analyze this data. Explain how this data informs antibiotic regimen optimization.
d. Experimental Design and Limitations: Discuss the critical factors to consider when designing experiments for each model, potential margins of error, utility, key assumptions and inherent limitations.
3. Phases of PK/PD Biofilm Model Experiments: Provide a distinct discussion on the three phases of the PK/PD biofilm model experiment (batch phase, continuous phase, and PK/PD phase).
4. Comparative Table of Models: Create a table that compares all the models described in the manuscript. This table should clearly outline the utility, advantages, and limitations of each model.
Author Response
We thank the reviewer for their comment. Please find the detailed point-by-point response attached.

Reviewer 2 Report
Comments and Suggestions for Authors
The authors have done a good job in organizing the different models and stating their limitations. However, the authors did not talk about the differences between the Sorbarod and the CDC biofilm infection model. Authors could also include which model is preferred. In my opinion, the authors could provide a brief explanation about the significance of measuring galactomannan release. Additionally, a few typos are found, e.g. line 225.
Author Response
Reviewer 2
Comments: The authors have done a good job in organizing the different models and stating their limitations. However, the authors did not talk about the differences between the Sorbarod and the CDC biofilm infection model. Authors could also include which model is preferred. In my opinion, the authors could provide a brief explanation about the significance of measuring galactomannan release. Additionally, a few typos are found, e.g. line 225.
- The difference between the sorbarod and the CDC Biofilm model is now discussed as “The major differences between the CDC reactor and the sorbarod model lie in two key aspects. First, the CDC reactor allows for the study of the time-course antibiofilm effects of antimicrobials by enabling the sampling of coupons at multiple time points throughout the experiment. In contrast, the sorbarod model permits sampling only at the end of the experiment, which requires dismantling the setup. Second, the structure of the biofilm in the sorbarod model may be heterogeneous due to the physical scale and irregularity of the cellulose fibre, whereas the CDC reactor typically provides more uniform conditions for biofilm formation”(org/10.3390/pathogens2020288, doi:10.1016/j.jmb.2015.09.002, doi:10.1111/jam.15200.) on the revised manuscript on line 348-355. In addition, the differences in the use, advantages and limitations of the two models is described briefly on the revised manuscript on Table 1 on page 14.
- The significance of measuring the galactomannan release is now discussed as “The level of galactomannan released is used as a surrogate measure of the viable fungal cells and hence the anti-fungal activity of the agents being investigated; a progressive decrease in the level of this biomarker correlates with fungicidal activity” on the revised manuscript on line 454-457.
Reviewer 3 Report
Comments and Suggestions for Authors
Static in vitro time-kill studies are not representative of clinical studies except when continuous infusion is used. Therefore, they are inadequate to predict optimal intermittent dosing regimens in individual patients unless a PK/PD model is developed that can predict the effects of different concentration-time profiles. However, dynamic in vitro models allow for the monitoring of killing kinetics, drug concentrations, and resistance development over several days because they allow drug concentrations to be continuously adjusted to mimic the in vivo PK profile. Previous studies in this area are insufficient to confirm the clinical significance of PK/PD relationships observed in vitro. In this case, the broader application of dynamic in vitro infection models in antimicrobial optimization is relatively limited. When the presented review is examined in detail, it is determined that it extensively discusses model setup, mathematical simulation approaches, and the application of dynamic in vitro infection models used in antimicrobial development and optimization. In addition, this study attempts to elucidate the relationships between dynamic in vitro infection models, pharmacokinetic data, and pharmacokinetic/pharmacodynamic models to guide the design of antimicrobial dosing regimens. It has been determined that the study presented in its current form has been prepared in an original and innovative way, written in an impressive language and contains very useful information to eliminate the deficiencies in this area. Therefore, I am of the opinion that it should be accepted in its presented form.
Author Response
Reviewer 3
Static in vitro time-kill studies are not representative of clinical studies except when continuous infusion is used. Therefore, they are inadequate to predict optimal intermittent dosing regimens in individual patients unless a PK/PD model is developed that can predict the effects of different concentration-time profiles. However, dynamic in vitro models allow for the monitoring of killing kinetics, drug concentrations, and resistance development over several days because they allow drug concentrations to be continuously adjusted to mimic the in vivo PK profile. Previous studies in this area are insufficient to confirm the clinical significance of PK/PD relationships observed in vitro. In this case, the broader application of dynamic in vitro infection models in antimicrobial optimization is relatively limited. When the presented review is examined in detail, it is determined that it extensively discusses model setup, mathematical simulation approaches, and the application of dynamic in vitro infection models used in antimicrobial development and optimization. In addition, this study attempts to elucidate the relationships between dynamic in vitro infection models, pharmacokinetic data, and pharmacokinetic/pharmacodynamic models to guide the design of antimicrobial dosing regimens. It has been determined that the study presented in its current form has been prepared in an original and innovative way, written in an impressive language and contains very useful information to eliminate the deficiencies in this area. Therefore, I am of the opinion that it should be accepted in its presented form.
We thank the reviewer for their thorough assessment and recommendations.